# Prevalence of Noncommunicable Disease (NCDs) risk factors in Tamil Nadu: Tamil Nadu STEPS Survey (TN STEPS), 2020

T. S. Selvavinayagam[1‡], Vidhya Viswanathan[1,2‡], Archana Ramalingam[2]*, Boopathi Kangusamy[2], Bency Joseph[2], Sudharshini Subramaniam[3], J. Sandhiya Sheela[2], Soniya Wills[2], Sabarinathan Ramasamy[2], Vettrichelvan Venkatasamy[2], Daniel Rajasekar[2], Govindhasamy Chinnasamy[2], Elavarasu Govindasamy[2], Augustine Duraisamy[2], D. Chokkalingam[2], Dinesh Durairajan[2], Mosoniro Kriina[2], Harshavardhini Vasu[4], Jerard Maria Selvam[5], Uma Sakthivel[4], Prabhdeep Kaur[2], Senthilkumar Palaniandi[6]

**1** Directorate of Public Health and Preventive Medicine, Government of Tamil Nadu, Chennai, India, **2** ICMR-National Institute of Epidemiology, Chennai, Tamil Nadu, India, **3** Institute of Community Medicine, Madras Medical College, Chennai, India, **4** Tamil Nadu Health Systems Reforms Project, Government of Tamil Nadu, Chennai, India, **5** National Health Mission, Tamil Nadu, Chennai, India, **6** Health and Family Welfare Department, Government of Tamil Nadu, Chennai, India

‡ TSS and VV are joint first authors on this work.
* rarchana@nieicmr.org.in

**Data Availability Statement:** The data underlying the results presented in the study are available

## Abstract

### Background

Noncommunicable diseases (NCDs) account for nearly 75% of all deaths in Tamil Nadu. The government of Tamil Nadu has initiated several strategies to control NCDs under the Tamil Nadu Health Systems Reform Program (TNHSRP). We aimed to estimate the prevalence of NCD risk factors and determine the predictors of diabetes and hypertension, which will be helpful for planning and serve as a baseline for evaluating the impact of interventions.

### Methods

A state-wide representative cross-sectional study was conducted among 18-69-year-old adults in Tamil Nadu in 2020. The study used a multi-stage sampling method to select the calculated sample size of 5780. We adapted the study tools based on WHO's STEPS surveillance methodology. We collected information about sociodemographic factors, NCD risk factors and measured blood pressure and fasting capillary blood glucose. The predictors of diabetes and hypertension were calculated using generalised linear models with 95% confidence intervals (95% CI).

### Results

Due to the COVID-19 pandemic lockdown, we could cover 68% (n = 3800) of the intended sample size. Among the eligible individuals surveyed (n = 4128), we had a response rate of 92%. The mean age of the study participants was 42.8 years, and 51% were women. Current tobacco use was prevalent in 40% (95% CI: 33.7–40.0) of men and 7.9% (95% CI: 6.4–

from Mendeley Data at URL DOI: 10.17632/vt86jmcdz3.1.

**Funding:** Fund Received Author: Dr. Prabdeep kaur Grant Number: G.O.(D).No.352 Full name of funder: Tamil Nadu Health System Reform Program (TNHSRP) URL of funder website: https://tnhsp.org/tnhsrp/ The funders had no role in study design, data collection and analysis, decision to publish, or preparation of the manuscript.

**Competing interests:** The authors have declared that no competing interests exist.

9.8) of women. Current consumption of alcohol was prevalent among 39.1% (95% CI: 36.4–42.0) of men. Nearly 28.5% (95% CI: 26.7–30.4) of the study participants were overweight, and 11.4% (95% CI: 10.1–12.7) were obese. The prevalence of hypertension was 33.9% (95% CI: 32.0–35.8), and that of diabetes was 17.6% (95% CI: 16.1–19.2). Older age, men, and obesity were independently associated with diabetes and hypertension.

## Conclusion

The burden of NCD risk factors like tobacco use, and alcohol use were high among men in the state of Tamil Nadu. The prevalence of other risk factors like physical inactivity, raised blood pressure and raised blood glucose were also high in the state. The state should further emphasise measures that reduce the burden of NCD risk factors. Policy-based and health system-based interventions to control NCDs must be a high priority for the state.

## Introduction

Noncommunicable diseases (NCDs) pose a significant global health challenge, accounting for 74% of deaths worldwide, per WHO [1]. NCDs account for over 65% of all deaths in India, according to the Global Burden of Disease Report, 2019 [2]. Though the epidemiological transition to a predominance of NCDs happened as early as 2003, it has not been uniform for all states of India [3]. India is a vast and diverse nation, with each state exhibiting significant variations in socio-economic conditions, culture, and levels of development. These differences can profoundly impact the prevalence of diseases and health outcomes across different states. Tamil Nadu, a southern state with a population of 72.1 million, is among the earlier states to undergo epidemiological transition. In 2019, more than 75% of all deaths in Tamil Nadu were due to NCDs, and high blood pressure alone led to nearly 22% of all deaths [2].

The government of Tamil Nadu has initiated various interventions to control NCDs [4]. The state is implementing the National Programme for Prevention and Control of Cancer, Diabetes, Cardiovascular Diseases and Stroke (NPCDCS), which includes screening, diagnosis, treatment and follow-up services for Diabetes, Hypertension, Cervical Cancer and Breast Cancer [5]. The World Bank funded the "Tamil Nadu Health System Reform Program" (TNHSRP) in 2019 to strengthen the institutional mechanisms and improve the management of NCDs [6].

Most NCDs are attributed to behavioural risk factors like unhealthy diet, physical inactivity, tobacco and alcohol use and biological risk factors like high blood glucose, high blood pressure, high cholesterol, and high body mass index. There are few studies conducted in different parts of Tamil Nadu that suggest a high prevalence of NCD risk factors. A study from Chennai City describes how tobacco use remains a public health issue, with a prevalence rate of 28.3%. Another study highlighted a concerning trend of overweight and obesity among adults, revealing a prevalence rate of 52.4%. The prevalence of Diabetes mellitus was 32.4%, and that of hypertension was 38.8% in the study conducted in Coimbatore [7, 8]. A nationally representative survey for India was done in 2017–18 to estimate the national-level prevalence of NCD risk factors [9]. Several Indian states have conducted state-wide STEPS surveys to generate locally relevant baseline estimates [10–13]. However, no such state-wide representative study for NCD risk factors has been done in Tamil Nadu. Estimating the burden of these risk factors

through the World Health Organization's (WHO) standard STEPS survey methodology will guide public health action for preventing NCDs [14].

With this background, we conducted a representative survey using the WHO STEP-wise approach to estimate the burden of risk factors for NCDs in Tamil Nadu in 2020. We also determined the predictors of diabetes and hypertension among the adult population of Tamil Nadu in 2020.

## Materials and methods

### Study setting and population

Tamil Nadu has a population of 72.1 million and a total area of 130,060 square km. Nearly 48.4% lived in urban regions, while 51.6% lived in rural areas [15]. We included adults aged 18 and 69 who stayed in the selected households for over six months. The standard WHO NCD STEPS Survey methods indicate including risk factor surveys for adults aged 18 and 69. Therefore, we included the same age group to ensure comparability with other studies. We excluded pregnant women, post-partum women (up to 42 days post-delivery), and anyone who could not respond to the interview owing to serious illness. We did the preparatory work for the study during the latter half of 2019 and began the data collection in January 2020.

### Sample size

We calculated the sample size by assuming the prevalence of NCD risk factors to be 50% with 95% confidence intervals and an allowable margin of error of 5%. We also adjusted for the complex multi-stage sampling design using a design effect of 1.5. Using the above-mentioned parameters, we arrived at a sample size of 578. Since we wanted precise age-gender specific estimates for the study outcomes, we adjusted the sample size calculations for four age groups (18–29, 30–44, 45–59 and 60–69 years) and both genders (male and female). Therefore, the total sample size required was 4624. After adjusting for 20% non-response, the total sample size was 5780. We decided to cover the sample size by sampling 30 adults from 193 clusters (merged villages and wards).

### Sampling strategy

We used a multi-stage cluster sampling method for the survey. In stage 1, from the sampling frame of all villages and wards (wards refer to administrative subdivisions within a city or municipality in India) in Tamil Nadu, we selected 193 clusters using the probability proportional to size (households) systematic sampling method (PPSS). We selected one census enumeration block (CEB) from each cluster using the probability proportional to size (PPS) method in stage 2. We enumerated the total number of households in the selected CEBs. In stage 3, we selected thirty households from each CEB using a systematic random sampling technique.

In stage 4, we employed the KISH method, a technique commonly used in survey research for randomly selecting participants. This method involves selecting participants based on a predetermined pattern, such as choosing every nth person from a list or grid of potential participants. Specifically, we selected one individual between the ages of 18 to 69 from each selected household. To facilitate the sampling process, we utilised sampling software developed by ICMR-NIE, which helped us efficiently select the appropriate sampling units at each stage of our study.

## Data collection tools

We collected the data using a questionnaire based on the WHO STEP-wise surveillance approach, which was locally adapted [16]. We translated the questionnaire to the local language (Tamil) and back-translated it to ensure that the meaning and intent of the questions were maintained. The STEPS survey had three different stages, as described below:

In STEP I, we collected information about sociodemographic details, behavioural risk factors like tobacco use, alcohol use, dietary habits, physical activity, and history of diabetes and hypertension through a questionnaire. We also asked about health-seeking behaviour for diabetes and hypertension.

In STEP II, the trained data collectors used standard methods to measure anthropometric parameters like height, weight, and waist circumference [16, 17] We used the SECA 213 stand-alone for measuring the height and the SECA 803 battery-operated electronic weighing scale for measuring the weight. The waist circumference was measured using SECA 201 ergonomic retractable tape. Blood pressure was recorded thrice using a calibrated professional digital BP apparatus (Omron 1300 HBP) among all the participants.

In STEP III, we measured the capillary fasting blood glucose after ensuring a minimum fasting period of 8 hours (ACCU-CHEK Instant S).

## Data collection procedure

We recruited 10 data collection teams (each with four investigators and one supervisor) and trained them for two weeks. During the first week, Q by Q (Question by Question) training was done for the data collectors. Standard methods of measuring study parameters such as anthropometry (height, weight, waist circumference), blood pressure, and capillary blood glucose measurements were taught. The data collectors were trained in using the ODK app for mapping villages, enumerating primary sampling units, entering data, and taking consent. During the second week of the training, mock sites were chosen, and a few sample/dummy participants were selected to observe and oversee the field-level work. Each trained team collected the data for 19–20 clusters and covered each cluster in about 4–5 days. The teams used tablets for data collection using an Android-based open data kit (ODK) app.

In each selected CEB, the team members listed all the households and uploaded them to the central server. The required 30 households were selected based on systematic random sampling using the built-in feature in the ODK platform. All eligible adults from each selected household were enumerated, and then the data was uploaded to the ODK app. The app automatically selected one eligible individual from each household using the KISH method.

The team members informed the selected participants about the survey and obtained their consent. They used the ODK app and administered the questionnaire to the participants. (The detailed questionnaire is added as a supplementary material S1 File). The trained staff also measured the anthropometric parameters and recorded the BP (three BP measurements were taken after ensuring a minimum of 5-minute intervals between them). The study participant was advised to maintain an 8-hour fasting state the following day. The next morning, the trained staff did a fasting capillary blood sugar test.

If the selected participant was unavailable for an interview (despite two house visits), refused to participate or was found ineligible due to pregnancy, they were included in the non-response list. Basic demographic details of the individuals in the non-response list were collected.

## Data management

The data was stored in password-protected servers with multi-layer security policies. We did daily data backup in an external storage device. The ODK app had sufficient in-built quality

checks to reduce data entry errors. A separate data quality team reviewed each team's data collected in 1–2 clusters and ensured appropriate data quality.

## Ethical considerations

Institutional Ethics Committee approval was obtained from Madras Medical College, Chennai. All participants were provided information regarding the survey using a participant information sheet, and informed consent was obtained. Confidentiality of the participants' identity was maintained by assigning a unique ID. People detected with Diabetes Mellitus and hypertension were referred to the nearest government health facility for confirmation of diagnosis and further management.

## Data analysis

The operational definitions that we had used for the key outcome variables as given in Table 1.

The data from the Android ODK app was backed up in the central server daily and exported to Excel. Data cleaning, coding and analysis were done using STATA SE (version 15.0) software (StataCorp LLC, Texas, USA). We reported proportions with 95% confidence intervals for categorical variables. We reported mean and 95% confidence intervals for continuous variables such as blood pressure, BMI and fasting blood glucose levels. We did a complex survey analysis to account for multistage-cluster-based sampling. We used generalised linear models to test the association between risk factors and outcomes like hypertension and diabetes and determined prevalence ratios (PR) and 95% CI. All the factors found to be associated based on univariable analysis were included in the multivariable generalised linear model. We checked for multicollinearity in the model using the Variation Inflation Factor (VIF) for all independent variables. We also determined the model's fitness using the likelihood ratio (lrtest) test of the full and nested models (S1 Table). We used a cut-off of p<0.05 to determine statistical significance. The independent predictors for other NCD risk factors like smoking, alcohol use, physical inactivity and obesity were also done and presented in the S2 Table.

**Table 1. Operational definitions of key outcome variables used in the study.**

| Outcome | Definitions |
|---|---|
| **Current tobacco use**. [16] | Use of any form of tobacco (smoke and/or smokeless) in the last 30 days preceding the survey |
| **Current alcohol use** [16] | Consumption of alcohol in the last 30 days preceding the survey |
| **Inadequate consumption of fruits and vegetables** [16] | Eating less than five servings of fruit and vegetables on an average per day (one serving is equivalent to 80–100 grams) |
| **Insufficient physical activity** [17] | Less than 150 minutes of moderate-intensity physical activity per week OR less than 75 minutes of vigorous-intensity physical activity per week OR an equivalent combination of moderate and vigorous-intensity physical activity accumulating< 600 MET minutes per week |
| **Overweight** [18] | Body Mass Index (BMI) $\geq 25.0 \leq 29.9$ kg/m2 |
| **Obesity**. [18] | BMI $\geq 30$ kg/m2 |
| **Central obesity**. [19] | A waist circumference of $\geq$90cm in males and $\geq$80cm in females |
| **Hypertension**. [20] | Anyone with **AT LEAST ONE** of the following: i. Average of second and third BP reading: Systolic Blood Pressure (SBP) $\geq$ 140 OR Diastolic Blood Pressure (DBP) $\geq$ 90 mmHg ii. Currently taking anti-hypertensive medication |
| **Diabetes** [21] | Anyone with **AT LEAST ONE** of the following: i. Fasting capillary Glucose $\geq$ 126 mg/dl ii. Currently taking hypoglycaemic drug (either oral drugs or insulin) |

## Results

### Sociodemographic characteristics of the study participants

We began the survey in January 2020 and identified 4128 eligible individuals by mid-March 2020. Among them, 3800 (92%) participated in STEP I and II and 3672 (89%) in STEP III. However, we could not proceed with the data collection due to the national lockdown in response to the COVID-19 pandemic. Overall, we covered around 68% (N = 3800) of the calculated sample size of 5780. The age and gender characteristics of responders and non-responders are given in the S3 Table.

The mean age of the study participants was 42.8 years, and 55.2% were in the age group of 18–44 years. Among the study participants, 51.1% were women. Nearly 20% of the study participants had a college degree, and 16.4% had no formal education. One-fourth (25.9%) of the participants were unskilled labourers / agricultural labourers, and another 26.1% were homemakers (Table 2).

### Prevalence of behavioural risk factors

The overall prevalence of current smoking was 12.5% (95% CI: 11.3–13.8). Women did not report smoking, and the prevalence among men was 25.6% (95% CI: 23.3–27.9) (Table 3). The mean age of starting to smoke among current smokers was 21.1 years, and the mean duration of smoking was 22.6 years (Table 4). Current tobacco use was prevalent in 40% (95% CI: 37.3–42.7) of men and 7.9% (95% CI: 6.4–9.8) of women (Table 3).

Current consumption of alcohol was prevalent among 39.1% (95% CI:36.4–42) of men (Table 3). Among current alcohol users, the average number of drinks consumed during one drinking occasion was 6.5 standard drinks among men, whereas it was 4.5 among women (Table 4).

Physical activity was inadequate among 21.1% (95% CI: 19.3–23.4) of men and 10.5% (95% CI: 8.9–12.3) of women (Table 3). The mean number of servings of fruits and vegetables consumed per day was 3.4 servings and did not differ among men and women (Table 4).

**Table 2. Characteristics of the study participants of the Tamil Nadu STEPS Survey, 2020 (N = 3800).**

| Variables | Men (N = 1858) n (%) | Women (N = 1942) n (%) | Overall (N = 3800) n (%) |
|---|---|---|---|
| **Age groups** | | | |
| 18–44 | 1083 (58.3) | 1013 (52.2) | 2096 (55.2) |
| 45–69 | 775 (41.7) | 929 (47.8) | 1704 (44.8) |
| **Education** | | | |
| Illiterate | 163 (8.8) | 459 (23.6) | 622 (16.4) |
| Less than Class 6 | 337 (18.1) | 400 (20.6) | 737 (19.4) |
| Class 6 or 7 or 8 completed | 303 (16.3) | 288 (14.8) | 591 (15.6) |
| High School or Higher Secondary (Class 9 to 12) | 610 (32.9) | 470 (24.2) | 1080 (28.4) |
| Graduation /diploma /post-graduation | 438 (23.6) | 308 (15.9) | 746 (19.6) |
| No response | 7 (0.4) | 17 (0.9) | 24 (0.6) |
| **Employment Status** | | | |
| Professional/Medium to large business/ Middle to Senior Executives | 259 (13.9) | 84 (4.3) | 343 (9.0) |
| Agricultural landowner | 148 (8.0) | 66 (3.4) | 214 (5.6) |
| Sales and Marketing executives/ Clerical/self-employed & small business | 301 (16.2) | 94 (4.8) | 395 (10.4) |
| Skilled manual labourer | 362 (19.5) | 94 (4.8) | 456 (12.1) |
| Unskilled manual/agricultural labourer | 514 (27.7) | 470 (24.2) | 984 (25.9) |
| Homemaker | 14(0.8) | 976 (50.3) | 990 (26.1) |
| Others | 260 (14.0) | 158 (8.1) | 418 (11.0) |

**Table 3. Prevalence of noncommunicable disease risk factors among the study participants of the Tamil Nadu STEPS Survey, 2020 (N = 3800).**

| Variables | Age groups (in years) | Men (N = 1858) | | Women (N = 1942) | | Overall (N = 3800) | |
|---|---|---|---|---|---|---|---|
| | | n | % (95% CI) | n | % (95% CI) | n | % (95% CI) |
| Current smokers | 18–44 | 259 | 23.9 (20.9–27.2) | 0 | NA* | 259 | 12.4 (10.7–14.2) |
| | 45–69 | 216 | 27.9 (24.8–31.2) | 0 | NA* | 216 | 12.7 (11.2–14.4) |
| | 18–69 | 475 | 25.6 (23.3–27.9) | 0 | NA* | 475 | 12.5 (11.3–13.8) |
| Current smokeless tobacco use | 18–44 | 182 | 16.8 (14.6–19.3) | 31 | 3.1 (2.1–4.4) | 213 | 10.2 (8.8–11.7) |
| | 45–69 | 148 | 19.1 (15.7–23.0) | 123 | 13.2 (10.7–16.3) | 271 | 15.9 (13.5–18.6) |
| | 18–69 | 330 | 17.8 (15.6–20.1) | 154 | 7.9 (6.4–9.8) | 484 | 12.7 (11.2–14.4) |
| Current tobacco use | 18–44 | 398 | 36.8 (33.7–40) | 31 | 3.1 (2.1–4.4) | 429 | 20.5 (18.5–22.6) |
| | 45–69 | 345 | 44.5 (40.6–48.5) | 123 | 13.2 (10.7–16.3) | 468 | 27.5 (24.8–30.4) |
| | 18–69 | 743 | 40 (37.3–42.7) | 154 | 7.9 (6.4–9.8) | 897 | 23.6 (21.8–25.6) |
| Current alcohol use | 18–44 | 440 | 40.6 (37.3–44.1) | 3 | 0.3 (0.1–1.3) | 86 | 21.1 (19.3–33.1) |
| | 45–69 | 287 | 37 (33.6–40.7) | 2 | 0.2 (0.1–0.9) | 289 | 17 (15.2–18.9) |
| | 18–69 | 727 | 39.1 (36.4–42.0) | 5 | 0.3 (0.1–0.7) | 732 | 19.3 (17.8–20.8) |
| Physical inactivity (not meeting WHO recommendation) | 18–44 | 214 | 19.8 (17.2–22.6) | 97 | 9.6 (7.7–11.9) | 311 | 14.8 (13.0–16.9) |
| | 45–69 | 178 | 23 (20.0–26.2) | 107 | 11.5 (9.3–14.1) | 285 | 16.7 (14.9–18.8) |
| | 18–69 | 392 | 21.1 (19.3–23.4) | 204 | 10.5 (8.9–12.3) | 596 | 15.7 (14.2–17.3) |
| Overweight (BMI is > = 25 and < = 29.9) | 18–44 | 310 | 28.6 (25.7–31.7) | 260 | 25.7 (23.0–28.5) | 570 | 27.2 (25.1–29.4) |
| | 45–69 | 205 | 26.5 (23.2–30.0) | 308 | 33.2 (30.2–36.3) | 513 | 30.1 (27.6–32.8) |
| | 18–69 | 515 | 27.7 (25.4–30.2) | 568 | 29.3 (27.2–31.4) | 1083 | 28.5 (26.7–30.4) |
| Obese (BMI is > = 30) | 18–44 | 83 | 7.7 (6.1–9.6) | 153 | 15.1 (12.7–17.9) | 236 | 11.3 (9.8–12.9) |
| | 45–69 | 47 | 6.1 (4.6–8.0) | 147 | 15.8 (13.2–18.9) | 194 | 11.4 (9.7–13.3) |
| | 18–69 | 130 | 7.0 (5.8–8.4) | 300 | 15.5 (13.4–17.7) | 430 | 11.3 (10.1–12.7) |
| Central obesity | 18–44 | 333 | 30.8 (27.6–34.1) | 469 | 46.3 (42.3–50.3) | 802 | 38.3 (35.4–41.2) |
| | 45–69 | 308 | 39.7 (35.7–44.0) | 577 | 62.1 (58.4–65.7) | 885 | 51.9 (48.7–55.2) |
| | 18–69 | 641 | 34.5 (31.8–37.3) | 1046 | 53.9 (50.6–57.1) | 1687 | 44.4 (41.9–47) |
| Raised blood pressure | 18–44 | 341 | 31.5 (28.4–34.8) | 115 | 11.4 (9.3–13.9) | 456 | 21.8 (19.7–24.0) |
| | 45–69 | 385 | 49.7 (46.1–53.3) | 446 | 48 (44.8–51.2) | 831 | 48.8 (46.2–51.3) |
| | 18–69 | 726 | 39.1 (36.6–41.7) | 561 | 28.9 (26.7–31.2) | 1287 | 33.9 (32.0–35.8) |
| Raised blood sugar (n = 3672) | 18–44 | 129 | 12.4 (10.5–14.5) | 65 | 6.6 (5.1–8.4) | 194 | 9.6 (8.3–11.0) |
| | 45–69 | 205 | 27.3 (24.1–30.7) | 247 | 27.8 (24.4–31.2) | 452 | 27.5 (25–30.1) |
| | 18–69 | 335 | 18.7 (16.7–20.7) | 312 | 16.7 (14.8–18.7) | 646 | 17.6 (16.1–19.2) |

## Prevalence of biological risk factors

Nearly 28.5% (95% CI: 26.7–30.4) of the study participants were overweight, and 11.3% (95% CI: 10.1–12.7) were obese. Among women aged 45–69, 33.2% were overweight, and 15.8% were obese (Table 3). The mean BMI among men was 23.5 kg/m$^2$, whereas, among women, it was 24.8 kg/m$^2$ (Table 4). The prevalence of central obesity was 53.9% among women and 34.5% among men (Table 3). The mean waist circumference among men was 84.5 cm, and among women was 81.4 cm (Table 4).

The prevalence of hypertension was 33.9% (95% CI: 32.0–35.8) among the study participants. The prevalence of hypertension was higher in the 45–69 year age group (48.8%) as compared to those aged 18–44 years (21.8%) [Table 3]. The mean systolic and diastolic blood pressure was highest among men aged 45–69 years (136.8 mm Hg and 85.5 mm Hg, respectively) compared to all other age and gender groups [Table 4].

Diabetes was prevalent in 17.6% (95% CI: 16.1–19.2) of the study participants. Nearly 27.5% of those aged 45–69 years had diabetes compared to 9.6% among those aged 18–44 (Table 3). The study population had a mean fasting blood glucose level of 115 mg/dl. Fasting

**Table 4. Mean of noncommunicable disease risk factors among the study participants of the Tamil Nadu STEPS Survey, 2020 (N = 3800).**

| Variables | Age groups | Men (N = 1858) Mean (95% CI) | Women (N = 1942) Mean (95% CI) | Overall (N = 3800) Mean (95% CI) |
|---|---|---|---|---|
| Mean age of starting smoking | 18–44 | 20.2 (19.4 21.0) | | 20.2 (19.4–21.0) |
| | 45–69 | 22.1 (20.9–23.2) | | 22.1 (20.9–23.3) |
| | 18–69 | 21.1 (20.3–21.8) | | 21.1 (20.3–21.7) |
| Mean duration of smoking among current smokers | 18–44 | 14.4 (13.4–15.3) | | 14.4 (13.4–15.9) |
| | 45–69 | 32.4 (31.0–33.9) | | 32.4 (31.0–34.2) |
| | 18–69 | 22.6 (21.3–23.7) | | 22.6 (21.3–23.9) |
| Mean number of drinks consumed on average per drinking occasion among current drinkers | 18–44 | 6 (5.5–6.5) | 2 | 6 (5.5–6.5) |
| | 45–69 | 6.5 (5.6–7.3) | 4.5 (2.4–6.6) | 6.5 (6.0–7.3) |
| | 18–69 | 6.2 (5.7–6.7) | 3.3 (1.4–5.1) | 6.2 (5.7–6.7) |
| Mean number of servings of fruits and vegetables on average per day | 18–44 | 3.6 (3.2–3.9) | 3.5 (3.2–3.8) | 3.6 (3.3–3.9) |
| | 45–69 | 3.3 (3.1–3.7) | 3.4 (3.1–3.7) | 3.4 (3.1–3.6) |
| | 18–69 | 3.4 (3.2–3.8) | 3.5 (3.2–3.8) | 3.4 (3.2–3.8) |
| Waist circumference in cm | 18–44 | 83.3 (82.4–84.2) | 78.8 (77.7–79.9) | 81.2 (80.3–82.0) |
| | 45–69 | 86.2 (85.2–87.3) | 84.3 (82.3–86.3) | 85.2 (83.9–86.4) |
| | 18–69 | 84.5 (83.8–85.3) | 81.4 (80.1–82.7) | 83 (82.1–83.8) |
| BMI in kg/m2 | 18–44 | 23.7 (23.2–24.1) | 24.4 (23.9–24.9) | 24 (23.7–24.4) |
| | 45–69 | 23.4 (23.1–23.7) | 25.2 (24.8–25.7) | 24.4 (24.1–24.7) |
| | 18–69 | 23.5 (23.2–23.9) | 24.8 (24.2–25.2) | 24.2 (23.9–24.5) |
| Systolic BP in mm Hg | 18–44 | 128.9 (127.7–130.0) | 116.7 (115.7–117.7) | 123 (122.1–123.9) |
| | 45–69 | 136.8 (135.1–137.3) | 137.3 (135.8–138.8) | 137.1 (135.9–138.3) |
| | 18–69 | 132.2 (131.2–133.3) | 126.6 (125.4–127.7) | 129.3 (128.4–130.2) |
| Diastolic BP in mm Hg | 18–44 | 83.3 (82.6–84.1) | 75.4 (74.6–76.1) | 79.5 (78.9–80.0) |
| | 45–69 | 85.5 (84.5–86.5) | 81.9 (81.1–82.7) | 83.5 (82.9–84.2) |
| | 18–69 | 84.2 (83.6–84.9) | 78.5 (77.9–79.1) | 81.3 (80.8–81.8) |
| Fasting Capillary Glucose in mg/dl (n = 3672) | 18–44 | 112.8 (110.4–115.2) | 103.9 (102.0–105.6) | 108.4 (106.8–110.0) |
| | 45–69 | 125.8 (121.6–130.0) | 123.6 (119.2–127.9) | 124.6 (121.2–128.0) |
| | 18–69 | 118.3 (115.8–120.7) | 113.2 (110.7–115.5) | 115.3 (113.7–117.6) |

glucose levels were higher among those aged 45–69 years (124.6 mg/dl) as compared to those aged 18–44 years (108.4 mg/dl) [Table 4].

## Predictors of hypertension among the study participants

In univariable analysis, older age group (PR: 2.2), men (PR: 1.4), current smokers (PR: 1.23), current consumers of alcohol (PR: 1.5), those with inadequate physical activity (PR: 1.2), with central obesity (PR: 1.81), those who were overweight (PR: 1.6), obese (PR: 1.9) and those with diabetes (PR: 2.1) had a higher chance of having hypertension (Table 5). After adjusting for age and gender, current smoking (PR adjusted for age and gender = 1.0) was not associated with hypertension (Table 5).

The factors (Adjusted PR with 95% CI) associated with hypertension after adjusting for all variables were age group 45–69 years [2.2 (95% CI: 1.9–2.4)], being male [1.3 (95% CI: 1.2–1.5)], current consumer of alcohol [1.4 (95% CI: 1.3–1.6)], overweight [1.6 (95% CI:1.4–1.7)]

**Table 5. Predictors of hypertension in the study participants of the Tamil Nadu STEPS Survey, 2020 (N = 3672).**

| Characteristic | | Hypertension | | Unadjusted Prevalence Ratio (PR) | Adjusted Prevalence ratio * | Adjusted Prevalence Ratio (aPR) † |
|---|---|---|---|---|---|---|
| | | Yes | No | | | |
| | | n (%) | n (%) | (95% CI) | (95% CI) | (95% CI) |
| Age group | 45–69 yrs | 802 (48.8) | 841 (51.2) | 2.2 (2.0–2.5) | | 2.2 (1.9–2.4) |
| | 18–44 yrs | 434 (21.4) | 1595 (78.6) | Reference | | Reference |
| Gender | Male | 695 (38.7) | 1099 (61.3) | 1.4 (1.2–1.5) | | 1.3 (1.2–1.5) |
| | Female | 541 (28.8) | 1337 (71.2) | Reference | | Reference |
| Current smoking | Yes | 184 (40.4) | 271 (59.6) | 1.2 (1.1–1.4) | 1.0 (0.9–1.2) | 1.0 (0.9–1.1) |
| | No | 1052 (32.7) | 2165 (67.3) | Reference | Reference | Reference |
| Current alcohol consumption | Yes | 319 (45.1) | 386 (54.9) | 1.5 (1.3–1.6) | 1.3 (1.2–1.5) | 1.4 (1.3–1.6) |
| | No | 917 (30.9) | 2047 (69.1) | Reference | Reference | Reference |
| Uses Added salt (always) | Yes | 120 (33.6) | 237 (66.4) | 1 (0.9–1.2) | 1.0 (0.9–0.2) | 1.02 (0.9–1.2) |
| | No | 1116 (33.7) | 2199 (66.3) | Reference | Reference | Reference |
| Physical activity | Inadequate | 229 (39.6) | 350 (60.1) | 1.2 (1.1–1.3) | 1.0 (1.0–1.2) | 1 (0.9–1.1) |
| | Adequate | 1007 (32.6) | 2086 (67.4) | Reference | Reference | Reference |
| Waist circumference | ≥80cm in females & ≥90cms in males | 729 (45.0) | 890 (55.0) | 1.81 (1.6–2.0) | 1.8 (1.6–2.0) | 1.4 (1.2–1.6) |
| | <80cm in females & <90cm in males | 507 (24.7) | 1546 (75.3) | Reference | Reference | Reference |
| BMI | ≥30 kg/m2 | 202 (49.2) | 209 (50.9) | 1.9 (1.6–2.1) | 2.0 (1.8–2.3) | 1.6 (1.4–1.7) |
| | 25–29.9 g/m2 | 449 (43.2) | 591 (56.8) | 1.6 (1.5–1.8) | 1.6 (1.5–1.8) | 1.9 (1.7–2.2) |
| | Upto 24.9kg/m2 | 585 (26.3) | 1636 (73.7) | Reference | Reference | Reference |
| H/O diabetes | Yes | 377 (58.4) | 269 (41.6) | 2.1 (1.9–2.2) | 1.7 (1.6–1.8) | 1.5 (1.4–1.6) |
| | No | 859 (28.4) | 2167 (71.6) | Reference | Reference | Reference |

*Adjusted for age and gender

† Adjusted for all factors significant in univariable analysis and key confounders based on literature

or obese 1.9 (95% CI: 1.7–2.2)]. The people with diabetes had a 1.5 times higher prevalence of hypertension (95% CI of aPR: 1.4–1.6) than those without diabetes. It was adjusted for all factors significant in univariable analysis and key confounders based on literature (Table 5).

## Predictors of diabetes among the study participants

In univariable analysis, older people (PR: 2.9, 95% CI: 2.5–3.3), those with inadequate physical activity (PR: 1.5, 95% CI: 1.2–1.8), with central obesity (PR: 2.6, 95% CI: 2.3–3.0), those who

**Table 6. Predictors of diabetes in the study participants of the Tamil Nadu STEPS Survey, 2020, N = 3672.**

| Characteristic | | Diabetes | | Unadjusted Prevalence Ratio (PR) | Adjusted Prevalence ratio * | Adjusted Prevalence Ratio (aPR) † |
|---|---|---|---|---|---|---|
| | | Yes | No | | | |
| | | n (%) | n (%) | (95% CI) | (95% CI) | (95% CI) |
| Age group | 45-69yrs | 194 (9.6) | 1835 (90.4) | 2.9 (2.5–3.3) | | 2.4 (2.1–2.8) |
| | 18-44yrs | 452 (27.5) | 1191 (72.5) | Reference | | Reference |
| Gender | Male | 334 (18.6) | 1460 (81.4) | 1.1 (1.0–1.3) | | 1.3 (1.1–1.6) |
| | Female | 312 (16.6) | 1566 (83.4) | Reference | | Reference |
| Current smoking | Yes | 71 (15.6) | 384 (84.4) | 0.9 (0.7–1.1) | 0.8 (0.6–0.97) | 0.9 (0.7–1.1) |
| | No | 575 (17.9) | 2642 (82.1) | Reference | Reference | Reference |
| Current alcohol consumption (30 days) | Yes | 116 (16.4) | 592 (83.6) | 0.9 (0.7–1.1) | 0.8 (0.7–1.1) | 0.9 (0.7–1.1) |
| | No | 530 (17.9) | 2434 (82.1) | Reference | Reference | Reference |
| Uses Added salt (always) | Yes | 54 (15.1) | 303 (84.9) | 0.8 (0.7–1.1) | 0.9 (0.7–1.1) | 0.9 (0.7–1.1) |
| | No | 592 (17.9) | 2723 (82.1) | Reference | Reference | Reference |
| Physical activity (WHO recommended) | Inadequate | 139 (24.0) | 440 (76.0) | 1.5 (1.2–1.8) | 1.4 (1.1–1.7) | 1.2 (1.1–1.5) |
| | Adequate | 507 (16.4) | 2586 (83.6) | Reference | Reference | Reference |
| Waist circumference | ≥80cm in females & ≥90cms in males | 435 (26.9) | 1184 (73.1) | 2.6 (2.3–3.0) | 2.5 (2.2–2.9) | 1.7 (1.4–2.2) |
| | <80cm in females & <90cm in males | 211 (10.3) | 1842 (89.7) | Reference | Reference | Reference |
| BMI | ≥30 kg/m2 | 109 (26.5) | 302 (73.5) | 2.2 (1.8–2.6) | 2.1 (1.8–2.4) | 1.8 (1.6–2.1) |
| | 25–29.9 kg/m2 | 267 (25.7) | 773 (74.3) | 2.1 (1.8–2.5) | 2.3 (1.9–2.8) | 1.9 (1.6–2.4) |
| | Up to 24.9kg/m2 | 270 (12.2) | 1951 (87.8) | Reference | Reference | Reference |
| H/O Hypertension | Yes | 377 (30.5) | 859 (69.5) | 2.8 (2.4–3.2) | 2.1 (1.8–2.5) | 1.9 (1.6–2.2) |
| | No | 269 (11.0) | 2167 (89.0) | Reference | Reference | Reference |

*Adjusted for age and gender

† Adjusted for all factors significant in univariable analysis and key confounders based on literature

were overweight (PR: 2.1, 95% CI: 1.8–2.5)), obese (PR: 2.2, 95% CI: 1.8–2.6) and those with hypertension (PR: 2.8, 95% CI: 2.4–3.2) had a higher chance of having diabetes (Table 6). Prevalence ratios did not differ when adjusted only for age and gender and when adjusted for all factors significant in univariable analysis and key confounders based on literature (Table 6).

The factors (Adjusted PR with 95% CI) associated with diabetes after adjusting for all variables were individuals aged 45–69 years [2.4 (95% CI: 2.1–2.8)], men [1.3 (95% CI: 1.1–1.6)], overweight [1.9 (95% CI: 1.6–2.1)] and obesity [1.8 (95% CI: 1.6–2.4) Individuals with

hypertension had a 1.9 (95% CI: 1.6–2.2) times higher prevalence of diabetes than those without diabetes (Table 6). Predictors of other NCD risk factors, such as smoking, alcohol, physical activity and abdominal obesity, are given in S3 Table.

## Clustering of risk factors among the study participants

Overall, 27.9% of the study participants had three or more risk factors for noncommunicable diseases. Among the men aged 45–69 years, nearly 40% had three or more risk factors. Among the women aged 18–44, almost 45% had no risk factor for noncommunicable disease (S1 Fig).

## Discussion

We documented a high prevalence of noncommunicable disease risk factors in the first-ever representative survey in the southern state of Tamil Nadu. Most of the risk factors were more prevalent among males, especially in the age group of 45–69. Obesity and central obesity were more prevalent among women. One-third of the population had hypertension, and nearly one in six had diabetes. Tamil Nadu is one of the few states with published data on the burden of risk factors based on the WHO STEPS methodology. The other states/union territories (UTs) with state-level representative estimates are Punjab, Haryana, Kerala, Madhya Pradesh and Puducherry [10–13].

The high prevalence of NCD risk factors in the state can be explained by the epidemiological transition that has happened in the state. The Global burden of disease estimates between 1990 and 2019 show that the top causes of morbidity and mortality have shifted to a predominance of NCDs [2]. This could be explained by the rapid urbansiation and globalisation that has been happening in the state.

Men tend to have a higher prevalence of alcohol and smoking as compared to women, and this may be due to social and cultural norms and peer pressure [22]. The high prevalence of smoking and alcohol use among men requires urgent attention. The prevalence of current smoking among men in Tamil Nadu was comparable to the national estimates based on the National Noncommunicable Diseases Monitoring Survey (NNMS) done in 2017–18 (23%) and the National Mental Health Survey (NHMS) done in 2015–16 (25%) [9, 23]. The Global Adult Tobacco Survey (GATS-2) done in 2016–17 reported a slightly lower prevalence of current smoking in Tamil Nadu (21%) [24]. The difference could be due to the methodology adopted by the two studies.

The low prevalence of current alcohol use among women (0.3%) was similar to various other surveys in India [9, 10, 12]. However, the prevalence of current alcohol use among men in the survey (39.1%) was higher compared to other states which used a similar methodology and definitions to report alcohol use, like Kerala (28.9%), Punjab (27.4%), Haryana (18.8%), and the national average reported by NNMS survey (28.3%) [9, 10, 12, 13].

Studies from other LMICs also show that smoking and alcohol are more prevalent among men than women [25–27]. Among men, the prevalence of smoking was 30%, 47%, and 28% in Sri Lanka, Bangladesh, and Nepal respectively [25–27]. The prevalence of alcohol use among men was 43% and 39% in Sri Lanka and Nepal respectively [25, 27].

The high prevalence of tobacco and alcohol use necessitates urgent interventions. The WHO has recommended cost-effective interventions for tobacco use and alcohol use in the form of MPOWER and SAFER strategies [28, 29]. MPOWER employs six key and effective strategies to combat the global tobacco epidemic, which include: 1) Monitoring tobacco consumption and the effectiveness of preventive measures; 2) Protect people from tobacco smoke; 3) Offering help to quit tobacco use; 4) Warn about the dangers of tobacco; 5) Enforce bans on tobacco advertising, promotion and sponsorship; and 6) Raise taxes on tobacco [28]. SAFER

focuses on cost-effective methods to reduce and prevent alcohol-related harm. These methods include 1) Strengthen restrictions on alcohol availability, 2) Enforce drunk driving counter-measures, 3) Facilitate access to screening, brief interventions, and treatment, 4) Enforce bans or comprehensive restrictions on alcohol advertising, sponsorship, and promotion, 5) Raise prices on alcohol through excise taxes and pricing policies [29]. The state must raise taxes on tobacco and alcohol products while restricting access to the availability of both. Strict enforcement of rules against drunk driving and comprehensive restrictions on alcohol and tobacco advertising, sponsorship and promotions must be done. Providing access to screening, interventions and treatment for alcohol and tobacco abuse at primary care will significantly help those who wish to quit and their families.

The prevalence of overweight among the individuals in Tamil Nadu (28.5%) is consistent with that seen in the NNMS survey (26.1%), Kerala (30%), and Punjab (28.6%) STEPS surveys [9, 10, 13]. However, the prevalence of obesity in Tamil Nadu (11.3%) is much higher than the national average (6.2%) [9]. The only state with a higher prevalence is Punjab (12.8%), whereas other states like Kerala (8.9%), Haryana (9,4%) and Madya Pradesh have a lower prevalence of obesity (3.4%) compared to Tamil Nadu [10, 12, 13, 30]. The prevalence of obesity in other LMICs like Sri Lanka, Bangladesh, and Nepal was found to be 11%, 5%, and 4.3%, respectively [25–27]. The high prevalence of overweight and obesity calls for urgent interventions. Reducing the intake of an unhealthy diet and improving physical activity is easier said than done. Providing conducive environments which would make healthier choices more accessible for the population is needed. Making fruits and vegetables accessible through free coupons provided by the public distribution system (PDS) can help improve the intake of fruits and vegetables. Community gyms, walkways and dedicated cycling paths can help improve physical activity.

Tamil Nadu has a high burden of hypertension and diabetes, putting the population at a high risk of cardiovascular disease-related mortality. The prevalence of hypertension in Tamil Nadu (33.9%) is similar to Puducherry (33.6%) and Kerala (30.4%) [10, 11]. However, it is higher than the national average (28.5%) [9]. Only Punjab has a higher prevalence of hypertension (40.1%) than Tamil Nadu, whereas other states like Haryana (26.2%) and Madya Pradesh have a lower prevalence (22.3%) [12, 13, 30]. The prevalence of hypertension in other LMICs like Bangladesh, Nepal, and Pakistan were 21%, 25% and 40%, respectively [26, 27, 31].

The prevalence of diabetes in Tamil Nadu (17.6%) is higher than the national average of 9.3% [9]. The burden is similar to Kerala (19.2%) and Haryana (15.5%) [10, 13]. The prevalence of Diabetes in other LMICs like Sri Lanka, Pakistan, Bangladesh, and Nepal are 15%, 16%, 8%, and 6% respectively [25–27, 31].

Older age, male gender and being overweight were major risk factors for hypertension and diabetes burden. These results align with findings from other studies [32–38]. The pathophysiological relationship between diabetes/hypertension with older age, physical inactivity and obesity is well established. With the rapid development and urbanisation in the state, the lifestyle and behaviours of individuals have changed. Maintaining a healthy body weight will reduce the burden of hypertension and diabetes. The state should prioritise older individuals and men for service provision for early detection and management.

## Strengths and limitations

We used the standardised STEPwise survey approach proposed by WHO, which is one of the strengths of the survey. We surveyed a representative population through a well-designed sampling strategy; hence, our results are internally valid for Tamil Nadu. We used standard calibrated instruments for anthropometric and blood pressure measurements to reduce information bias in the study.

Our study is not devoid of limitations. We could not cover the intended sample size due to the national lockdown for the COVID-19 pandemic. Therefore, we revised our analysis plan and presented the prevalence estimates of NCD risk factors for both genders in two age groups instead of four. Even though we had covered only 68% of the sample size, the prevalence estimates of the critical outcome variables have narrow confidence intervals, indicating the adequacy of the sample size. Another limitation to consider is the social desirability bias, which may affect the responses related to socially sensitive issues like tobacco use. Our estimates for physical inactivity were lower than national estimates and surveys from similar states. GPAQ requires participants to have a certain literacy level and depends on leisure time physical activity, limiting its use in LMICs like India [17]. This might have affected the estimates of physical inactivity in the current study. However, we used the questionnaire to ensure the comparability of results with other STEPS surveys. The factors associated with diabetes and hypertension should be interpreted cautiously as data about outcomes and exposures were collected simultaneously.

## Conclusions and recommendations

In conclusion, more than one-fourth of the adults aged 18–69 years in Tamil Nadu have a clustering of three or more risk factors for NCD. The burden of behavioural risk factors like the consumption of alcohol and tobacco is high among men in the state. Intake of fruits and vegetables is inadequate among most of the adult population. Tamil Nadu has a higher burden of obesity, diabetes and hypertension than most states in India where STEPS surveys have been done. The interventions to reduce the burden of NCD risk factors should be a high priority for the state. Health promotion campaigns should be launched through various media to educate people about the importance of healthy lifestyles. A combination of policy-based and health system-based interventions should be considered. The policy-based options include (i) raising tobacco and alcohol taxes, (ii) enforcing strict measures against advertising alcohol and tobacco products, (iii) creating parks, walkways and open spaces for physical activity and (iv) coupons for fruits and vegetables in the public distribution system. Screening must be done for all adults above 30 years at public and private health facilities and workplaces to detect diabetes and hypertension. Strengthening the primary health system to detect, treat, track, and follow up individuals with diabetes and hypertension is the need of the hour for the state of Tamil Nadu.

## Supporting information

**S1 Fig. Clustering of risk factors for noncommunicable diseases by age and gender, TN STEPS Survey 2020.**
(TIF)

**S1 Table. Multicollinearity tests between independent variables included in the glm model for hypertension diabetes.**
(DOCX)

**S2 Table. Predictors of NCD risk factors among study participants of the Tamil Nadu STEPS Survey, 2020.**
(DOCX)

**S3 Table. Responders and non-responders of the TN STEPS Survey, 2020 by age and gender.**
(DOCX)

**S1 File. STEPS survey for NCDs in Tamil Nadu, 2020 questionnaire.**
(DOC)

## Acknowledgments

This study was done in collaboration with the Tamil Nadu Health System Reform Program (TNHSRP) and the Institute of Community Medicine, Madras Medical College, Chennai. We thank the Directorate of Public Health and Preventive Medicine, Government of Tamil Nadu. We are also grateful to our colleague Mr P Kamaraj (Senior Technical Officer, ICMR-NIE) for his insightful discussions and valuable feedback throughout the project. His contributions greatly enhanced the quality of our work. We extend our appreciation to the field team for the smooth conduct of the study. We also thank the participants who volunteered their time and provided us with the data needed for this study.

## Author Contributions

**Conceptualization:** T. S. Selvavinayagam, Vidhya Viswanathan, Archana Ramalingam, Bency Joseph, Sudharshini Subramaniam, J. Sandhiya Sheela, Harshavardhini Vasu, Jerard Maria Selvam, Uma Sakthivel, Prabhdeep Kaur.

**Data curation:** T. S. Selvavinayagam, Vidhya Viswanathan, J. Sandhiya Sheela, Soniya Wills, Sabarinathan Ramasamy, Daniel Rajasekar, Govindhasamy Chinnasamy, Elavarasu Govindasamy, Augustine Duraisamy, D. Chokkalingam, Dinesh Durairajan, Mosoniro Kriina.

**Formal analysis:** Vidhya Viswanathan, Archana Ramalingam, Boopathi Kangusamy, Soniya Wills, Sabarinathan Ramasamy, Vettrichelvan Venkatasamy, Daniel Rajasekar, Govindhasamy Chinnasamy, Elavarasu Govindasamy, Augustine Duraisamy, D. Chokkalingam, Dinesh Durairajan, Mosoniro Kriina, Prabhdeep Kaur.

**Funding acquisition:** Prabhdeep Kaur.

**Investigation:** J. Sandhiya Sheela.

**Methodology:** Vidhya Viswanathan, Archana Ramalingam, Boopathi Kangusamy, Bency Joseph, Sudharshini Subramaniam, J. Sandhiya Sheela, Vettrichelvan Venkatasamy, Prabhdeep Kaur.

**Project administration:** T. S. Selvavinayagam, Vidhya Viswanathan, Archana Ramalingam, Bency Joseph, Vettrichelvan Venkatasamy, Daniel Rajasekar, Uma Sakthivel, Prabhdeep Kaur, Senthilkumar Palaniandi.

**Software:** Sabarinathan Ramasamy, Vettrichelvan Venkatasamy.

**Supervision:** Archana Ramalingam, Boopathi Kangusamy, Sudharshini Subramaniam, Sabarinathan Ramasamy, Harshavardhini Vasu, Jerard Maria Selvam, Uma Sakthivel, Prabhdeep Kaur, Senthilkumar Palaniandi.

**Validation:** Sabarinathan Ramasamy.

**Visualization:** Archana Ramalingam, Bency Joseph.

**Writing – original draft:** Vidhya Viswanathan, Archana Ramalingam.

**Writing – review & editing:** T. S. Selvavinayagam, Vidhya Viswanathan, Archana Ramalingam, Boopathi Kangusamy, Bency Joseph, Sudharshini Subramaniam, J. Sandhiya Sheela, Soniya Wills, Sabarinathan Ramasamy, Vettrichelvan Venkatasamy, Daniel Rajasekar, Govindhasamy Chinnasamy, Elavarasu Govindasamy, Augustine Duraisamy, D.

Chokkalingam, Dinesh Durairajan, Mosoniro Kriina, Harshavardhini Vasu, Jerard Maria Selvam, Uma Sakthivel, Prabhdeep Kaur, Senthilkumar Palaniandi.

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
