## [Decision Letter · Decision Letter 0]

18 Jun 2023

PONE-D-23-15618Prevalence of Noncommunicable Disease (NCDs) Risk Factors in Tamil Nadu: Tamil Nadu STEPS Survey (TN STEPS), 2020.PLOS ONE

Dear Dr. Ramalingam,

Thank you for submitting your manuscript to PLOS ONE. After careful consideration, we feel that it has merit but does not fully meet PLOS ONE’s publication criteria as it currently stands. Therefore, we invite you to submit a revised version of the manuscript that addresses the points raised during the review process.

 Under the comments from Reviewer 1, the comment on not selecting population aged 70 years and above could be due to the STEPs guidelines that recommends covering up to 69 years, so consider that comment accordingly.  Under the comments from Reviewer 2, the comment about response rate would not require so much attention. But follow the comment from the Academic editor about sample size calculation which is a must to be addressed before proceeding further. 

We look forward to receiving your revised manuscript.

Kind regards,

Krishna Kumar Aryal

Academic Editor

PLOS ONE

Journal Requirements:

Additional Editor Comments:

Archana Ramalingam et al, a good manuscript on STEPS Survey. Please consider following additional points to those of reviewers.

In the abstract, rather than a comparable way of presenting the conclusions, it would be ideal to showcase what are the major ones and how big it is in your state and what factors are primarily associated with, and drawing the conclusion in that line would be more appropriate. The conclusion in the main body of the manuscript seems more logical, try to align the conclusion of abstract as well to that.

Methods: What does it mean by abruptly stopped the survey, at what stage it was stopped and what happened after that?

Sample size.

There is a calculation fallacy in the sample size calculation.

The initial sample size of 578 is fine with the parameters and the value used. Then in the next step it has to be multiplied by 8 (to cover 8 age sex groups as per the description), which means the total sample will be 4624. This also looks fine. Now where is the adjustment for 80% response rate (assumption of 20% non-response). I am sure authors are aware about the response rate adjustment. The required sample size for your survey as per the design is 4624, now you will have to increase the sample size by adjusting the non-response which will lead to 5780. If you calculate the same using Steps Sample size calculator you will get 5762 as final sample size with your description. That minimal difference of 5762 and 5780 doesn’t make a difference, but the missing adjustment of non-response assumption is a complete fallacy. However, when I see the results, you have only used 3 age groups in analysis. Please see this thoroughly and update the manuscript accordingly. There is a serious mismatch here.

Reviewers' comments:

Reviewer's Responses to Questions

**Comments to the Author**

1. Is the manuscript technically sound, and do the data support the conclusions?

Reviewer #1: Partly

Reviewer #2: Yes

2. Has the statistical analysis been performed appropriately and rigorously? 

Reviewer #1: Yes

Reviewer #2: Yes

3. Have the authors made all data underlying the findings in their manuscript fully available?

Reviewer #1: No

Reviewer #2: Yes

4. Is the manuscript presented in an intelligible fashion and written in standard English?

Reviewer #1: Yes

Reviewer #2: Yes

5. Review Comments to the Author

Reviewer #1: Dear Editor,

Thank you for the opportunity to review the manuscript entitled “Prevalence of Noncommunicable Disease (NCDs) Risk Factors in Tamil Nadu: Tamil Nadu STEPS Survey (TN STEPS), 2020.”. I read the manuscript with great interest and found it highly relevant. However, I have the following suggestions for further improvement.

Abstract:

1. Study aim(s) is/are missing in the abstract.

2. Methods line 14: The authors mentioned calculating adjusted prevalence ratios of diabetes and hypertension. Reading the introduction, I had an impression that several NCDs risk factors will be assessed. The results also provide prevalence estimates for several factors. So the statistical analysis under the method in the abstract needs revisiting and being comprehensive to include other factors as well.

Introduction

3. Literature review could be improved by providing information on the prevalence of NCDs risk factors at the local level. I understand that no previous state-wide studies have been conducted, but I wonder if there are small-scale local studies that could provide some estimates of NCDs risk factor prevalence in Tamil Nadu.

4. A brief statement on why existing nationwide data is insufficient may provide context for readers to understand inter-state diversity in India. In other words, why is it important to get estimates at the state level when national-level data exists? The authors should understand that readers may not be familiar with the heterogeneity (i.e., socio-economy, culture, and development) between states in India that could potentially impact the prevalence of diseases and health outcomes.

5. Why were predictors of only two factors, i.e., diabetes and hypertension, determined? I believe the authors have sound justification for doing that, but that justification is not articulated for readers.

Methods

6. Study Setting and Population: Why those 70 and above were excluded from participation? Please provide your justification.

7. Sampling strategy: International readers may not be familiar with concepts like ward, so I suggest adding a brief context for comprehension. Likewise, KISH method is not self-explanatory, so a brief explanation will help the reader to understand how sampling was done.

8. Data collection procedure: please add what the two-week training for data collection teams included.

9. Lines 85-87 are confusing and contradictory. The authors mentioned, “The team members enumerated all the eligible individuals in the selected households.” The following line then indicated, “Using the KISH technique, the app had an in-built feature to select one participant per household automatically.” I am confused if data was collected from everybody or a single person in the household.

10. Data analysis: For the regression models, was multicollinearity tested and how the model fit was determined?

11. The operational definition of variables in Table 1 is very helpful, but it would be helpful to know what was asked to participants and/or how each was measured. The method section should have a section on variables measurements, but if that’s not feasible, it should be in the supplemental text or table.

Discussion

12. The discussion section lacks possible explanations for observed findings. For example, what explains the high burden of NCDs risk factors overall, and what mechanism supports differences between males and females?

Minor comment:

13. The manuscript needs proofreading for grammar issues. For example: in several places, there are inappropriate cases. Abbreviation (e.g., Govt of TN) is used without specifying at first use.

Reviewer #2: Thank you for inviting me to review this interesting manuscript.

The manuscript provides an overview of a study conducted in Tamil Nadu, India, to estimate the prevalence of noncommunicable diseases (NCDs) and associated risk factors.

Abstract

Overall, the manuscript abstract effectively presents the background, methods, key findings, and implications of the study.

Please add the relevant numbers and statistics (95% confidence intervals etc.) to support your statements in the results section of the abstract

Methods: A representative cross-sectional study was conducted among 18-69-year-old adults in 10 Tamil Nadu in 2020- representative in terms of national, provincial or what?

Main body:

Introduction:

Paragraph 1- Please shows the global landscape of NCD highlighting its prevalence, morbidity, and mortality on a worldwide scale and narrow your focus to India, exploring the specific challenges and implications faced by the country in relation to prevalence of noncommunicable diseases (NCDs) and associated risk factors.

Results

It is important to acknowledge that you covered 82% of the intended sample size of 4624 participants, with a final sample size of 3800 individuals. However, it is crucial to address the potential implications of the reduced sample size on the generalizability and statistical power of your findings. Consider discussing any potential biases that may arise from the incomplete coverage of the intended sample and any steps taken to mitigate such biases.

What was the response rate? I couldn’t locate where the actual response rate is presented. Please also clarify how it was calculated.

Please add 95% CI before range of result, check result section for e.g line 153

Discussion

The comparisons with studies undertaken in other countries are useful. Please compare your results with those from other studies conducted outside India. It seems that the discussion section is missing the discussion of variables that reached significance in the multivariable analysis for the purpose of comparing them to studies conducted outside India.

MPOWER and SAFER strategies what are they? – please check and provide full form of all abbreviated form in initial throughout the manuscript.

6. PLOS authors have the option to publish the peer review history of their article (what does this mean?). If published, this will include your full peer review and any attached files.

Reviewer #1: No

Reviewer #2: **Yes: **Anil Poudyal

---

## [Author Response · Author response to Decision Letter 0]

10 Oct 2023

Response to Additional Editor's comment- Thank you for the comments. We have incorporated your suggestions into the revised manuscript.

We have attached a Word document "Responses to the comments" addressing all your specific comments and suggestions.

Response to Reviewer 1- Thank you for the comments. We have incorporated your suggestions into the revised manuscript.

We have attached a Word document "Responses to the comments" addressing all your specific comments and suggestions.

Response to Reviewer 2- Thank you for the comments. We have incorporated your suggestions into the revised manuscript.

We have attached a Word document "Responses to the comments" addressing all your specific comments and suggestions.

---

## [Decision Letter · Decision Letter 1]

7 Nov 2023

PONE-D-23-15618R1Prevalence of Noncommunicable Disease (NCDs) Risk Factors in Tamil Nadu: Tamil Nadu STEPS Survey (TN STEPS), 2020.PLOS ONE

Dear Dr. Ramalingam,

Thank you for submitting your manuscript to PLOS ONE. After careful consideration, we feel that it has merit but does not fully meet PLOS ONE’s publication criteria as it currently stands. Therefore, we invite you to submit a revised version of the manuscript that addresses the points raised during the review process.

We look forward to receiving your revised manuscript.

Kind regards,

Krishna Kumar Aryal, MPH, PhD

Academic Editor

PLOS ONE

Journal Requirements:

Additional Editor Comments:

Authors Archana Ramalingam et al have done a good attempt to respond to reviewers and editors to improve the quality of the paper. However, there still seem to be some issues not addressed and a new comment is also added below.

A new author has been added to the authors list. Please provide justification/clarification for the same.

The way CIs are mentioned appear to be done in quite a rush, please check it. I noticed in abstract but check throughout, it has to be mentioned as 95%CI: 36.4-42.0. You don'e need to write % in CI.

Line 97 of track changes version - ….had an overall prevalence of 32.44%.... please be consistent in writing digits after decimal. Here you have written 2 digits, but elsewhere single digit. I would suggest using single digit throughout.

The first para of the results section where you talk about response rate etc… you have not updated the information as per the comment on sample size. Authors appear to have not gone through the changes requested seriously. Please address in all sections pertaining to the comments.

Once you are done, please review the whole paper thoroughly for minor issues of copy edits, language, spelling and inconsistency. And I suggest you use a third person to review the language once you are done with all the edits.

Thank you, and if you could address these soon, the editors and journal office might help to take action on time as soon as the revised version is submitted.

All the best.

Reviewers' comments:

Reviewer's Responses to Questions

**Comments to the Author**

1. If the authors have adequately addressed your comments raised in a previous round of review and you feel that this manuscript is now acceptable for publication, you may indicate that here to bypass the “Comments to the Author” section, enter your conflict of interest statement in the “Confidential to Editor” section, and submit your "Accept" recommendation.

Reviewer #2: All comments have been addressed

2. Is the manuscript technically sound, and do the data support the conclusions?

Reviewer #2: Yes

3. Has the statistical analysis been performed appropriately and rigorously? 

Reviewer #2: Yes

4. Have the authors made all data underlying the findings in their manuscript fully available?

Reviewer #2: Yes

5. Is the manuscript presented in an intelligible fashion and written in standard English?

Reviewer #2: Yes

6. Review Comments to the Author

Reviewer #2: (No Response)

7. PLOS authors have the option to publish the peer review history of their article (what does this mean?). If published, this will include your full peer review and any attached files.

Reviewer #2: **Yes: **Anil Poudyal

---

## [Author Response · Author response to Decision Letter 1]

14 Jan 2024

Response to editor comments: We thank the editor for the comment. We have incorporated your suggestions into the revised version.

---

## [Editor Report · Decision Letter 2]

23 Jan 2024

Prevalence of Noncommunicable Disease (NCDs) Risk Factors in Tamil Nadu: Tamil Nadu STEPS Survey (TN STEPS), 2020.

PONE-D-23-15618R2

Dear Dr. Ramalingam,

We’re pleased to inform you that your manuscript has been judged scientifically suitable for publication and will be formally accepted for publication once it meets all outstanding technical requirements.

Kind regards,

Krishna Kumar Aryal, MPH, PhD

Academic Editor

PLOS ONE
---

## [Editor Report · Acceptance letter]

26 Apr 2024

PONE-D-23-15618R2 

PLOS ONE

Dear Dr. Ramalingam, 

I'm pleased to inform you that your manuscript has been deemed suitable for publication in PLOS ONE. Congratulations! Your manuscript is now being handed over to our production team.

Kind regards, 

on behalf of

Dr Krishna Kumar Aryal 

Academic Editor

PLOS ONE